# Determinants of Physical Activity and Dietary Habits among Adults in Ghana: A Cross-Sectional Study

**DOI:** 10.3390/ijerph19084671

**Published:** 2022-04-13

**Authors:** Kingsley Agyemang, Amrit Banstola, Subhash Pokhrel, Nana Anokye

**Affiliations:** Division of Global Public Health, Department of Health Sciences, College of Health, Medicine and Life Sciences, Brunel University London, Uxbridge UB8 3PH, UK; kingsley.agyemang@brunel.ac.uk (K.A.); amrit.banstola@brunel.ac.uk (A.B.); subhash.pokhrel@brunel.ac.uk (S.P.)

**Keywords:** obesity, non-communicable diseases (NCDs), physical activity, diet, Ghana

## Abstract

A critical understanding of the interrelationship between two behavioral decisions—participating in physical activity, and eating healthily—is lacking in Ghana. This study aimed to determine which factors affect each of the two behavioral decisions, jointly and separately, among adults aged 18 years or older in three metropolises (Kumasi, Accra, and Tamale) of Ghana. The data from the Ghana Obesity Survey 2021 were used. A bivariate probit model was fitted to estimate nonlinear models that indicate an individual’s joint decision to participate in physical activity and consume a healthy diet. A positive correlation (r = 0.085; *p* < 0.05) was found between these two decisions, indicating a relationship between these two behavioral decisions. The common correlates between these decisions were self-reported good health status, high income, and attitudes toward being overweight. Men were more likely to be physically active but less likely to eat well. Both religion and culture determined participation in physical activity, but not the consumption of a healthy diet. Marital status determined diet, but not physical activity. The new knowledge gained from this analysis around the nature and the extent of the interconnectedness between physical activity and diet is critical to devising targeted interventions for obesity prevention in Ghana.

## 1. Introduction

Obesity is a multifaceted health condition determined by the interaction of genetic, behavioral, physiological, psychological, and social factors [1]. It is a growing public health problem worldwide, with an estimated 1.9 billion adults being overweight and more than 650 million obese in 2016 [2]. It was the fourth leading risk for deaths and the fifth leading risk for disability-adjusted life years (DALYs) worldwide in 2019 [3]. The World Health Organization (WHO) estimates that the prevalence of obesity will continue to rise in all age groups over the next decade, including in low- and middle-income countries, where the numbers are increasing [4,5,6].

In Ghana, the numbers are alarming. The prevalence of overweight doubled, and of obesity tripled, between 1993 and 2014 [7]. In 2016, the prevalence of overweight and obesity was estimated to be 25.4% and 17.1%, respectively [8]; 43% of adults were either overweight or obese in the same year. Ethnicity is one of the important determinants of overweight and obesity. There is inequality in the prevalence of obesity among ethnic groups and tribes: obesity was highest in the Akan and Ga-Dangme ethnic groups [9]. Obesity increases the risk of several major non-communicable diseases (NCDs), including cardiovascular diseases, type 2 diabetes, asthma, and cancers [10]. A systematic review also showed that people with obesity have a 3.74-fold greater risk of developing COVID-19 than those without obesity [11]. In Ghana, 45.6% of adults with type 2 diabetes were either overweight or obese in 2016 [8].

Several unhealthy lifestyle behaviors—such as lack of physical activity, unhealthy diet, tobacco use, and harmful use of alcohol—are associated with an increased risk of obesity and NCDs. The importance of both physical activity and diet in preventing obesity is well documented [12]. However, the focus of interventions has predominantly been on changing individual behaviors, which is less effective and less efficient than intervening in more than one risk behavior simultaneously [13]. Addressing obesity and its behavioral determinants requires a holistic approach to understanding the critical arguments connected to energy balance—physical activity, and uptake of a healthy diet [14,15,16].

The commonly held belief that physical activity can “outrun” a bad diet has been challenged with new evidence [17]. Likewise, engaging in one healthy behavior (e.g., sports and exercise) could lead to the uptake of another healthy behavior (e.g., eating healthily), because people tend to transfer their knowledge and confidence gained in one behavior to another [18]. The implications of this hypothesis for public health are far-reaching—with well-designed strategies, people can be supported to complement one healthy behavior with the other for a balanced and healthy lifestyle. This further means that it is possible to achieve cumulative and positive effects of engaging in physical activity and a healthy diet on survival and quality of life [19]. Both of these “substitution” and “complementary” effects could also co-occur, and the occurrence of either effect could be a function of the ordering of behaviors [19]. Thus, depending on the type and degree of incentives (or perceived benefits associated with the behaviors), coupled with perceived outlays to realize those benefits, people may substitute or complement these health behaviors at a given time.

The economic theories underpinning human behavior and resulting decision making suggest that individual preferences and decisions to eat a healthy diet and participate in physical activity are influenced by various factors. These factors could be at both the population level (e.g., changes in the economic environment by price promotions to healthy foods, taxes on less healthy foods, or tax exemptions to promote physical activity) and the individual level (e.g., income, current health status) [20]. From a traditional (neo-classical) economic point of view, people are rational in their preferences and decide on their diet and exercise behavior to maximize their utility [21]. In contrast, behavioral economics focuses on how individuals make actual decisions based on the psychological underpinnings of human behavior, considering insights from various social and biological sciences to influence their choices and behaviors [22]. From a behavioral economics perspective, diet and physical activity choices are not always within people’s rational control, given the complex interactions of genetic, behavioral, physiological, psychological, and social factors in determining obesity [23].

This study focused on diet and physical activity, as these are the main focus of current policies to manage obesity [14,24]. A problem, however, is that the evidence base is sparse on the understanding of the nature and degree of interrelationship between these two behaviors—particularly in the ethnic groups in Ghana. Therefore, we sought to explore this interrelationship, assuming that individuals make these decisions subject to various constraints under which they have to operate.

## 2. Materials and Methods

### 2.1. Study Setting and Sampling

The study was conducted in Kumasi, Greater Accra, and Tamale—the country’s most populous metropolises and major economic hubs. These metropolises reflect the population, including the topography of the country. Over the years, they have attracted people across the geographical strata of the country, and all of the major ethnic groups can be found in these areas.

The minimum sample required per study site was estimated using the population size, with a 95% confidence interval and a 3% margin of error. In 2021, the population of Kumasi was 3,348,000, that of Accra was 2,514,000, and that of Tamale was 642,000. A total sample size of 3200 reflects the sum of the sample size of each study site (Kumasi 1067; Accra 1067; and Tamale 1066). The effective sample size was adjusted to 3840 participants, allowing for a 20% non-response rate.

A three-stage stratified sampling design was used in the survey (selecting enumeration areas, systematically selecting households, and randomly selecting individuals within households).

Stage one: 108 enumeration areas were selected using equal allocation, irrespective of the population size, from the three metropolises. A simple random sampling technique was used to identify these enumeration areas. The sampling frame was the list of enumeration areas from the 2010 Ghana Population and Housing Census.Stage two: 20 households in each enumeration area were randomly selected using the Ghana Statistical Service’s existing sampling frame to derive 2160 households.Stage three: All adults (aged 18 years or older) from each household were invited to participate in the survey.

### 2.2. Design and Validation of the Questionnaire

A structured questionnaire—i.e., the Ghana Obesity Survey 2021—was developed specifically for this study. The survey asked questions about participation in physical activity over the past four weeks. In addition, the survey explored the number of days spent undertaking physical activity, including duration and intensity of activity. The focus of this study was on the sports and exercise component of physical activity, as this represents a planned aspect often aimed at attaining health benefits [25,26] and, as such, can be relatively easily targeted by policies to reduce obesity. In addition, it is subject to less measurement error, since sports and exercise activities are usually premeditated and, hence, easier to recall by respondents [27]. Physical activity was defined as participation (or not) in sports, exercise, and other physical activities (captured through traditional activities in Ghana, e.g., shea nut picking, farming, fishing). Participants were asked, “Can you tell me if you have done any physical activity during the last four weeks, that is, since (date four weeks ago)?” Those responding “Yes” to this question were asked further questions about the name, duration, and intensity of the activity. Participants who provided the name of the physical activity, engaged for at least 15 min, and responded that the physical activity made them out of breath and sweaty were categorized as physically active. Those responding “No” to the question were classified as physically inactive.

The questionnaire also covered behavioral factors such as consumption of fruits and vegetables, sedentary behavior, smoking history, and alcohol consumption. Defining unhealthy eating was problematic, as nutritionally defined “balanced food” does contain some high-calorie, high-fat ingredients. Therefore, in-line with the WHO/FAO recommendations, a healthy diet was specified as eating at least five portions of fruits and vegetables, less than 5 g of iodized salt, and some nuts, whole grains, and legumes each day, along with the consumption of other elements of a composite diet for the day [28]. Participants were asked, “Thinking about the food you eat, I would like you to tell me how often you usually eat the following foods: fruit or vegetables, including fresh, frozen, dried, tinned, and pure fruit juice?” Those responding at least “once every day” to this question were categorized as having healthy dietary habits. Those responding to other options were classified as having unhealthy dietary habits.

This survey included questions on socio-demographic variables such as gender, age, marital status, ethnicity, religion, employment status, monthly household income, health status, area of residence, occupational status, disability, and length of stay at residence. The questionnaire also captured information about body mass index by measuring the height and weight of the participants using internationally standardized measurement scales. The questionnaire was adapted from Ghana’s validated national survey, i.e., the Ghana Statistical Service 2021 Population and Housing Census [29] and elsewhere [30,31,32].

### 2.3. Piloting

The questionnaire was pilot tested in March 2021 to identify any potential issues that could affect survey respondents and data collectors before being administered in the field [33]—for example, testing questionnaire duration, surveying procedures, willingness to participate, willingness and ability to answer questions, and the flow of the questions regarding the consistencies. Data collectors fed back this information at a pilot debrief, and the questionnaire was further revised before fieldwork. The main changes made were amending the named activities for sports and exercise to reflect the traditional activities in Ghana, and revising the questions on ethnicity and marital status to reflect major tribes and customary marriages, respectively.

### 2.4. Data Collection

Data collection was carried out by 54 trained field officers of the Ghana Statistical Service (GSS). There were nine teams; each team was made up of a supervisor and five enumerators. A three-day workshop was provided to them before data collection. Data were collected electronically on tablet computers in April 2021. To avoid respondent fatigue during the survey, we encouraged respondents to take their time and think about the answers. Respondents were informed that they were in full control of the survey and were allowed to take breaks during the survey. They were also informed that they could stop or withdraw from the study at any time, without giving a reason.

### 2.5. Missing Observations

The approach used to account for missing data was determined by the cause and proportion of missing data. The cause of missing data was identified by checking the applicability of the data collection to respondents—items “not applicable” (i.e., not relevant) or true missing (i.e., non-response). Where data were not applicable—for example, a question on frequency if a respondent reported no smoking—the most appropriate value was assigned (i.e., non-smoker or zero cigarettes smoked). Where data were genuinely missing, patterns of missing data were examined using descriptive statistics. Chi-squared and Fischer’s exact tests were used to check the association between the indicators of obesity (dependent variable) and dummy variables representing their item non-responses, so as to examine the mechanisms under which the missing data occurred (e.g., missing completely at random or not) [34]. Missing data were replaced using the mean-based imputation method. While this method is limited in reflecting the uncertainty around missing data points and reducing the variation in the dataset, this impact is expected to be minimal in this study, as the proportion of missing data was small (less than 5%) and, in such cases, the method for addressing missingness is inconsequential [35].

### 2.6. Analysis

Descriptive statistics were used to summarize the characteristics of the samples. A range of possible health behaviors in which people could be physically (in)active and adopt (un)healthy dietary habits was estimated using proportions.

A bivariate probit model [36] was used to translate the possibility of a “joint” decision-making process involved in an individual’s choice of physical activity and diet. A bivariate probit model is a parametric class of regression models. The joint distribution of the error terms associated with the choices (physical activity and healthy diet, in this case) is assumed to be a bivariate normal distribution. This model can thus be used to estimate the correlation between these joint choices and other factors, such as gender, age, marital status, ethnicity, religion, employment status, monthly household income, health status, area of residence, occupational status, disability, and length of stay at residence.

With the binary outcomes Yp (denoting physical activity) and Yh (indicating healthy eating), we considered:Yp = Xp β p + εp(1)
Yh = Xh β h + εh(2)
where Xp and Xh are correlates (independent variables) of physical activity and healthy diet choices, respectively. The assumption of independence of these two choices may be limiting, because one could argue that individuals know all possible scenarios in which they want to do various activities a priori, and gain a defined level of satisfaction from all of those activities. Therefore, Xp ≠ Xh and T > Ki (where T = total number of observations; Ki = the total number of independent variables), and
ε = [εp, εh]

We assumed, however, that the independent variables were strictly exogenous, giving us:E [ε | Xp, Xh] = 0

For any given equation, the error terms were not correlated across observations (σph IT), but were correlated across Equations (1) and (2). Therefore:E [εp εh | Xp, Xh] = σph IT

The null hypothesis (i.e., p equals zero) was that the estimation includes two probit models that can be estimated independently. This null hypothesis was tested via the rho and likelihood ratio (LR) statistics. In addition, marginal effects (MEs), estimated at sample mean values of independent variables, were computed for each variable. The threshold for statistical significance was set at ≤ 10% in all analyses because of the study’s exploratory nature. All analyses were performed using Stata 13 (StataCorp LLC, College Station, TX, USA).

## 3. Results

Of the 6000 cases issued, 3900 potential respondents were eligible for inclusion. Of these, 3348 respondents completed the survey (response rate of 86%). The majority of non-responses consisted of non-contact (*n* = 213)—for example, people having relocated. The average time taken to complete the survey was 27 min.

Table 1 shows that the average age of the respondents was 40 years, and the majority of them were women (60%). Most were employed (88%), had no disability (87%), were married or living with their partners (59%), and reported Christianity as their religion (55%). In addition, the majority reported favorable health (87%). The respondents were mainly Akans (38%) and Mole-Dagbani (34%). The majority had monthly personal income below GH¢ 1000 (77%) (GH¢ 6.27 = USD 1, as of 19 January 2022). The majority of the sample (71%) had lived at their current residence for over five years.

Percent of the respondents, 46% (*n* = 1520) did not engage in any healthy lifestyles, and only 10% (*n* = 341) were both physically active and adopted healthy dietary habits (Figure 1).

Table 2 shows the regression estimates for the joint estimation of physical activity and healthy diet. The results show that the correlation between participating in physical activity and eating a healthy diet was 0.085, which is statistically significant at less than 0.5%. This positive correlation indicates the existence of a relationship between these two behavioral decisions. The common factors in the decisions to participate in physical activity and eat a healthy diet were self-reported good health status, high income, and attitudes toward being overweight.

### 3.1. Physical Activity

Area of residence had the most significant association with physical activity. Compared with Kumasi residents, residents of the Tamale metropolis were 26% less likely to engage in physical activity. Another important factor was gender, with men being 17% more likely to be physically active. Religion had a mixed relationship with physical activity. While Muslims were more likely to be physically active (about 1% more) than Christians, individuals in other religions were 5% less likely. The tribe had a slightly higher but positive effect. Mole-Dagbanis and other minority tribes had a 7 to 9% higher probability of being physically active than Akans. Other variables with a positive effect included income and employment. High-income earners (>GH¢ 1000 per month) and the employed were 3–5% and 6% more likely to engage in physical activity, respectively. On the other hand, unfavorable health was negatively associated with physical activity (about 4% less likelihood). Regarding the effects of attitudes, about 83% (*n* = 5/6) of the indicators for attitude showed a significant but mixed correlation. Individuals who perceived being overweight as either a sign of beauty, a sign of good living, or hereditary were 4–6% less likely to be physically active. On the other hand, perceiving that obesity is unhealthy or caused by heaving eating was associated with an increased likelihood (4%) of being physically active.

### 3.2. Healthy Diet

Overall, the healthy diet equation had fewer predictors. Notably, it was neither associated with religion, culture, nor area of residence. Conversely, marital status was associated with a healthy diet, but not with physical activity. People who were not married were 3% less likely to eat healthily. Where a variable was significantly associated with both physical activity and a healthy diet, it showed the same direction of effect, though with varying levels of magnitude. The only exception was gender. While men were more likely to be physically active, they were less likely to eat healthily (around a 5% reduced probability). Unfavorable health was associated with a 3% decrement in the probability of eating healthily. Higher income had a positive effect (3%). Half of the attitudinal variables were significantly associated with healthy eating—underweight being hereditary (ME: −0.021), a sign of good living (ME: −0.003), and caused by insufficient exercise (ME: 0.021).

## 4. Discussion

This is the first study to explore joint estimators for examining the determinants of diet and physical activity specifically in sub-Saharan Africa. The most important finding of this study was that analyzing physical activity and eating behavior data separately—assuming that they are independent individual choices—may be biased. Joint estimators that account for the correlation between the residuals of both behaviors have been shown to correctly identify the dynamics of the functionality between the two sets of correlates. Although there is limited evidence in the obesity field to which to relate our findings, elsewhere in lifestyle behavior research, Anokye and Stamatakis [37] showed similar findings. They found joint estimation for sedentary behavior and physical activity to produce more efficient estimates than individual estimators.

The correlates of physical activity and diet were comparable but slightly dissimilar. Participation in physical activity was correlated with religion and culture, but not with consuming a healthy diet. Conversely, marital status determined healthy diet choices, but not physical activity. The evidence for correlation of marital status with physical activity is inconclusive. A systematic review on correlates of physical activity identified nine studies conducted with adults aged 18 years or older, and five of these studies were inconclusive, while others did not report on correlates [38]. However, an econometric estimation based on a bivariate probit model from a nationally representative sample in Portugal showed that being married was correlated with engaging in physical activity and eating a healthy diet [39].

In our study, men were more likely to be physically active but less likely to eat a healthy diet. Our study also showed that self-reported unfavorable health was negatively associated with physical activity, consistent with other studies [40,41]. A systematic review by Bauman et al. also reported that being male was correlated with physical activity [38]. A recent community-based cohort study that assessed correlates of physical activity among Sri Lankan adults (35 to 64 years old) found that being physically active was associated with being male and better self-reported health [40]. In this Sri Lankan study, self-reported health was based on a five-point Likert scale from “Very good” to “Poor” responses about their feeling about their general health. In a cross-sectional study utilizing data from a nationwide survey in Canada, self-reported health (categorized as excellent, good, fair, or poor) was associated with physical activity [41]. A systematic review of correlates of weight-related behaviors among adults (aged 18 years or older) in the Gulf States showed that being a woman was associated with low physical activity [42]. The relationship between men and physical activity may emanate from perceptions of physical activity and body image. For example, evidence from a current meta-analysis suggests that contemporary description of masculinity, male buoyancy, and general body image may feed into the narrative around participation in physical activity among men [43].

The relationship between male gender and the reduced likelihood of eating a healthy diet was similar to the findings of Saquib et al. [44]. A cross-sectional study from Bangladesh showed that Bangladeshi women (non-pregnant and aged over 30 years) preferred eating fruits and vegetables to fat-based foods, while men preferred the latter [44]. This difference could be linked to advertising companies that promote thinness as beauty among women and not men [45]. Furthermore, this difference may also be biologically induced, as demonstrated by Varì et al. [46]. Our study showed that people who were employed were engaged in physical activity. A study from Portugal showed that being unemployed did not increase the odds of participating in physical activity and eating a healthy diet among adults (those above 18 years old) [39]. A survey conducted in six rural communities in the United States showed that the employed ate healthier diets [47].

Another important finding of this study is the identification of the common determinants of physical activity and eating behavior. Those with positive attitudes toward being overweight, good health status, and higher household income were more likely to participate in physical activity and eat a healthy diet. One of the implications of this finding is that intervention to promote these behaviors needs to focus on people who have poor health status and lower household income. These are often people who have limited or no options to choose what to eat and/or whether to engage in physical activity. Unlike this finding, another study from Ghana showed that higher household income was associated with unhealthy lifestyle behaviors [48]. The association between income, physical activity, and eating behavior is complex, and is ultimately based on individual choices and decisions. Future research is required to understand why people with different household incomes make different decisions regarding their health, and will facilitate the design of effective interventions to overcome barriers to making healthy decisions.

### Strengths and Limitations

This study adds to the limited evidence examining the interrelationship between physical activity and diet. This is the first study to explore joint estimators for examining the determinants of diet and physical activity specifically in sub-Saharan Africa. This study used the data from the Ghana Obesity Survey, which is the first comprehensive lifestyle behavior survey in Ghana.

This study has some limitations. First, it was a cross-sectional study; therefore, it was not possible to examine causal inferences. A second limitation was the inclusion of study participants from three metropolitan areas representing the urban populations. The construct of the urban population is different from the rural population in Ghana. Therefore, future research needs to involve a mix of both urban and rural populations in Ghana, so as to provide a more comprehensive view of the relationships between physical activity and diet. Third, we relied on a single question based on fruit and vegetable intake to obtain data on dietary habits and determine dietary health. While fruit and vegetable intake information is vital in determining a healthy diet, it may not be adequate. A detailed questionnaire on dietary habits and food choices should be considered in future research. Finally, the operational definition of physical activity included participation in sports, exercise, and traditional household activities only. Physical activity is multifaceted, and includes various activities, such as sports and exercise, housework, occupational activity, and travel. It is important to note that physical activity and diet could also be influenced by other factors than those explored in this study, such as education, environment, price, health information, and social context. Others could argue for the assumption of physical activity and eating behavior as individual choices and decisions, because for people who are food insecure, there is little or no choice involved in deciding what type of food to eat, as they eat food just to survive [49].

## 5. Conclusions

The area of residence, being male, and having a good health status were associated with engagement in physical activity in Ghana. While engagement in physical activity was associated with religion and culture, none of these correlates was related to eating a healthy diet. Men were more likely to be physically active, but less likely to eat well. Conversely, marital status determined diet, but not physical activity. The common indicators of both behaviors included attitudes toward being overweight, good health status, and high income. The improved understanding of these nuances around how individuals from different population subgroups behave, in the context of the balanced need to be physically active and eat healthily, should guide health policymakers to devise targeted strategies to prevent and treat obesity.

## Figures and Tables

**Figure 1 ijerph-19-04671-f001:**
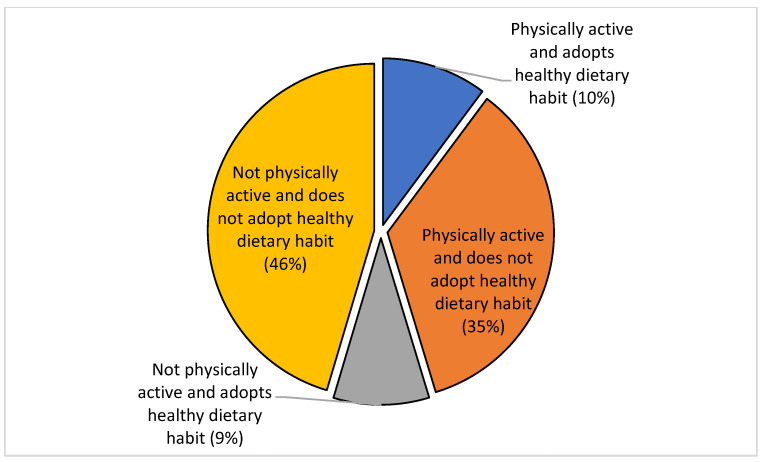
Participation in physical activity and adoption of healthy dietary habits.

**Table 1 ijerph-19-04671-t001:** Sociodemographic and economic characteristics of study participants (missing observations are indicated; *n* = 3348).

Variables	Number	Percent
**Annual personal income**		
<GH¢ 1000 *	2570	76.76
GH¢ 1000 to less than GH¢ 2000	613	18.31
GH¢ 2000 to less than GH¢ 3000	129	3.85
GH¢ 3000 to less than GH¢ 4000	23	0.69
GH¢ 4000 to less than GH¢ 5000	7	0.21
GH¢ 5000 and above	6	0.81
**Gender**		
Male	1328	39.67
Female	2019	60.30
Other	1	0.03
**Ethnicity**		
Akan	1268	37.87
Ga-Dangme	306	9.14
Ewe	145	4.33
Guan	17	0.51
Gurma	23	0.69
Mole-Dagbani	1149	34.32
Grusi	101	3.02
Mande	1	0.03
All others	338	10.10
**Marital status**		
Informal/living together	185	5.53
Married (civil/ordinance)	407	12.16
Married (customary/traditional)	394	11.77
Married (Islamic)	914	27.30
Married (other type)	62	1.85
Separated	129	3.85
Divorced	84	2.51
Widowed	228	6.81
Never married	945	28.23
**Religion**		
Catholic	225	6.72
Protestant	288	8.60
Pentecostal/Charismatic	1140	34.05
Other Christian	189	5.65
Islam	1393	41.61
Ahmadi	7	0.21
Traditionalist	17	0.51
No religion	79	2.36
Other (specify)	10	0.30
**Employment status**		
Employed	2944	87.93
Unemployed	404	12.07
**Disability**		
Yes	399	11.92
No	2931	87.54
Prefer not to say	18	0.54
**Health status**		
Very good	1330	39.73
Good	1602	47.85
Fair	340	10.16
Bad	69	2.06
Very bad	7	0.21
**Length of stay at current address**		
Less than six months	87	2.60
Six months to one year	91	2.72
One year to two years	226	6.75
Two to five years	564	16.85
More than five years	2380	71.09
**Region of residence**		
Kumasi	1202	35.90
Greater Accra	918	27.42
Tamale	1228	36.68

* GH¢ 6.27 = USD 1, as of 19 January 2022.

**Table 2 ijerph-19-04671-t002:** Joint estimation results for the probability of engaging in physical activity and eating healthily.

Independent Variables	Physical Activity		Healthy Diet	
	Coefficient	Marginal Effects	Coefficient	Marginal Effects
**Religion ^a^**				
Muslims	0.713 ***	0.008	−0.003	−0.002
Others	−0.789 ***	−0.052	0.167	0.034
**Ethnicity ^b^**				
Ga-Dangme	0.149	0.067	−0.144	−0.028
Ewe	0.139	0.063	−0.136	−0.026
Mole-Dagbani	0.236 *	0.093	−0.125	−0.033
Others	0.216 **	0.072	−0.015	−0.018
**Metropolis ^c^**				
Greater Accra	−0.176	−0.229	0.122	0.072
Tamale	−0.529 ***	−0.256	0.119	0.075
**Male ^d^**	0.474 ***	0.166	−0.101 *	−0.047
**Employed ^e^**	0.148 **	0.056	−0.074	−0.022
**Health status ^f^**				
Good	−0.123 **	−0.003	−0.290 ***	−0.031
Fair	−0.164 *	−0.037	−0.142	−0.008
Bad/very bad	−0.271	−0.061	−0.261	−0.018
**Have a disability? ^e^**	−0.077	−0.022	−0.025	−0.002
**Personal income ^g^**				
GH¢ 1000 to less than GH¢ 2000	0.235 ***	0.025	0.328 ***	0.027
GH¢ 2000 and above	0.464 ***	0.051	0.491 ***	0.023
**Marital status ^h^**				
Informal living/never married	−0.063	−0.001	−0.161 **	−0.017
Separated/divorced	−0.154	−0.027	−0.208 *	−0.018
Widowed	0.126	−0.011	0.334 **	0.039
**Age**	-0.002	−0.000	−0.004 *	−0.000
**Agree that “Being overweight is something you inherit from your parents”? ^e^**	−0.244 ***	−0.039	−0.287 ***	−0.021
**Agree that “Being overweight is a sign of good living”? ^e^**	−0.163 **	−0.040	−0.104 *	−0.003
**Agree that “Being overweight is a sign of beauty”? ^e^**	−0.138 **	−0.056	0.096	0.025
**Agree that “Being overweight is unhealthy”? ^e^**	0.148 **	0.043	0.041	0.005
**Agree that “Most people who are overweight have put on weight because they eat too much”? ^e^**	0.016	0.010	−0.035	−0.006
**Agree that “Most people who are overweight have put on weight because they exercise** **too little”? ^e^**	0.215 ***	0.038	0.264 ***	0.021
Constant	0.080		−0.493 ***	
Observations	3348			
Rho	0.085 **			

Significance levels: *** 1%; ** 5%; * 10%. Reference group: ^a^ Christians; ^b^ Akan; ^c^ Kumasi. ^d^ Female; ^e^ Yes; ^f^ very good; ^g^ < GH¢ 1000; ^h^ married.

## Data Availability

The data presented in this study are available upon request from the corresponding author.

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
