# Peer review of "Determinants of Physical Activity and Dietary Habits among Adults in Ghana: A Cross-Sectional Study"

_ijerph, 2022, doi:10.3390/ijerph19084671_

Round 1
Reviewer 1 Report
General comments:
The aim of the study was to determine which factors affect physical activity level and healthy diet intake, jointly and separately, using data from the Ghana Obesity Survey 2021. results indicated that both decisions are correlated.
Weakness: inconsistent referencing style.
Overall, reading of the article did not flow smoothly.
Specific comments:
Suggested edits or revisions are noted below:
Introduction:
Lines 29-30: Are these the most updated estimates?
Line 38: ... non-communicable diseases (NCDs).
Line 53: please replace "uptake" with "adoption' or "engaging in".
Lines 99-101: definition of "healthy diet" can be more specific.
Lines 103-105: "The questionnaire was pilot-tested in March 2021 to identify any potential ... before being administered in the field."
The Analysis section of the "Materials and Methods" is a bit too technical for a layman. Could the authors offer simpler explanation of the statistical procedures carried out?
Results: Table 1 headings to be revised; the age of respondents is confusing. It shows mean & SD on the left but its heading states "Percentage".
Table 2: How were the reference categories for the various variables chosen? Authors could include this in the Materials and Methods.
Healthy Diet: ME not previously defined? margin of error?
Author Response
Reviewer 1 comments:
Comment: Inconsistent referencing style.
Response: Thank you for pointing it out. Referencing style is now consistent for all references.
Comment: Overall, reading of the article did not flow smoothly.
Response: We have now improved the flow of the article by adjusting sentence structures.
Comment: Lines 29-30: Are these the most updated estimates?
Response: Updated estimate was available for the leading risk for deaths and disability-adjusted life-years from the recent Global Burden of Diseases (GBD) 2019 study. We have now used this recently available estimate in our article with appropriate reference.
Comment: Line 38: ... non-communicable diseases (NCDs).
Response: Non-Communicable Diseases (NCDs) is now written as non-communicable diseases (NCDs).
Comment: Line 53: please replace "uptake" with "adoption' or "engaging in".
Response: Thank you for suggesting these appropriate words. We have now replaced ‘uptake of’ with ‘engaging in’ in line 53.
Comment: Lines 99-101: definition of "healthy diet" can be more specific.
Response: We have now made the definition of a ‘healthy diet’ more specific on page 3, lines 131-134.
Comment: Lines 103-105: "The questionnaire was pilot-tested in March 2021 to identify any potential ... before being administered in the field."
Response: We have now amended the sentence as you have suggested. Please see page 3 (lines 149-150). Thank you.
Comment: The Analysis section of the "Materials and Methods" is a bit too technical for a layman. Could the authors offer simpler explanation of the statistical procedures carried out?
Response: We have now provided a simpler explanation of the statistical analysis used in the study. Please see the ‘Analysis’ section on page 4 and 5.
Comment: Results: Table 1 headings to be revised; the age of respondents is confusing. It shows mean & SD on the left but its heading states "Percentage".
Response: The age of respondents is now removed from the table as it is already stated in the text in line 222.
Comment: Table 2: How were the reference categories for the various variables chosen? Authors could include this in the Materials and Methods.
Response: This was an arbitrary choice. Variables with ‘zero’ while coding was chosen as the reference groups/categories. We did not specifically mention this in the article because we get the same results no matter what we choose. So, we have just noted the reference group as a footnote in the table.
Comment: Healthy Diet: ME not previously defined? margin of error?
Response: Thank you for pointing this out. ME refers to ‘Marginal Effects’. We have now included this in parenthesis in line 213 (where it first appeared in the article).
Reviewer 2 Report
The article Nexus between Physical Activity and Diet in Ghana studies the relationship between physical activity and diet, and socio-economic factors in Ghana. Although this kind of research may provide some useful data and shed light on some problems in the local community (as well Ghana as other similar countries), this study has some methodological deficiencies that may affect the quality of the data obtained in the study. Below are some of my major objections:
- Materials and methods section - there is lack of description of study group and the questionnaire. Although you describe a recrutation for the study very well, I consider it impossible to do assessment of methodological correctness where there are no description of the questionnaire (about what you asked? what was the scale in question? what was unsers? did the respondends unswer diferently than you assumed (e.g. have different answer then was it in the test)? And what was the study group? Did you asked about some major factors influencing diet and physical activity, such as weight and height (BMI) or particular work? There are ofcourse many more factors, those are only examples. If you did not, you cannot do any of conclusions about physical activity of respondends (for example - factory worker will not go run after work, but he have higher physical activity than the office worker who go play soccer 2 times per week for 45 minutes).
- in line 189 you wrote that 27 minutes was the mean time to fill the questionnaire - in the methods section, please describe how do you influence the respondends to eliminate the effect of fatigue while filling in the survey, which undoubtedly affects the quality of the data obtained.e.g. you have taken some breaks during the filling?
- in table 1 you describe different ethnicity of respondends. In my opinion you should carrefouly describe in introduction section how do they different between each other, why is it important to compare them.
- the nexus between physical activity and diet is basicly describe only in lines 261-267. In whole article you are not talking about it at all. Mostly you are describing how socio-economic factors are influencing physical activity, which is out of the aim of study in my opinion.
- how participants where qualified to physical active/inactive and healty/unhealthy diet (fig. 1)? you need to carrefouly describe it in method section. What was the questions that allow you to qualified them as one of those 4 options? Or was that just question, if there are physically active (or do you consider your diet as healthy) with yes/no answers?
Additionaly there are some editorial mistakes:
- lone 43 physical inactivity - i think it will be better when you write lack of physicla activity
- line 51-53 - sentence should be write differently, it is hard to understand now
- lines 65-69 - the references are in different style (Name&Name, 2020) than it s required by journal and in other paragraphs ([1]). Same in line 277
- table 1 - i belive that Authors wanted to use "<" sign in 3rd line, they use the opposite one
In my opinion, the article have to many flaws for current state. It should be at last rewritten and renamed, where more attention will be paid to material and methods section and more informations for the above will be given.
Author Response
Reviewer 2 comments:
Comment: Materials and methods section - there is lack of description of study group and the questionnaire. Although you describe a recruitment for the study very well, I consider it impossible to do assessment of methodological correctness where there are no description of the questionnaire (about what you asked? what was the scale in question? what was unsers? did the respondents answer differently than you assumed (e.g. have different answer then was it in the test)? And what was the study group? Did you asked about some major factors influencing diet and physical activity, such as weight and height (BMI) or particular work? There are of course many more factors, those are only examples. If you did not, you cannot do any of conclusions about physical activity of respondents (for example - factory worker will not go run after work, but he have higher physical activity than the office worker who go play soccer 2 times per week for 45 minutes).
Response: Thank you for your suggestion to improve the materials and methods section. This section has been substantially improved by taking into consideration your suggestions and feedback from other reviewers.
Comment: in line 189 you wrote that 27 minutes was the mean time to fill the questionnaire - in the methods section, please describe how do you influence the respondents to eliminate the effect of fatigue while filling in the survey, which undoubtedly affects the quality of the data obtained. e.g., you have taken some breaks during the filling?
Response: Thank you for this useful suggestion. We have described respondents’ fatigue in the ‘Data collection’ heading of the ‘Materials and Methods’ section in page 4 (lines 162-166).
Comment: in table 1 you describe different ethnicity of respondents. In my opinion you should carefully describe in introduction section how do they differ between each other, why is it important to compare them.
Response: Thank you for your suggestion to include information on ethnicity in the introduction. As suggested, we have now described the inequality in ethnicity and its importance in obesity research with links to physical activity and eating a healthy diet. Please see page 1 (lines 36-39) and page 2 (lines 83).
Comment: the nexus between physical activity and diet is basically describe only in lines 261-267. In whole article you are not talking about it at all. Mostly you are describing how socio-economic factors are influencing physical activity, which is out of the aim of study in my opinion.
Response: Thank you for raising this concern. The study aimed to explore the interrelationship between participating in physical activity and eating a healthy diet and assessing which factors affect each of them jointly and separately. In other words, this study sought to answer what is the relationship between physical activity and diet and are these lifestyle behaviours affected by the same variables? The discussion section has been amended to reflect this.
Comment: how participants where qualified to physical active/inactive and healthy/unhealthy diet (fig. 1)? you need to carefully describe it in method section. What was the questions that allow you to qualified them as one of those 4 options? Or was that just question, if they are physically active (or do you consider your diet as healthy) with yes/no answers?
Response: As suggested, we have now described how we categorized respondents as physically active/inactive on page 3 (lines 122-127) and eating a healthy/unhealthy diet on page 3 (lines 134-139).
Comment: line 43 physical inactivity - i think it will be better when you write lack of physicla activity
Response: Thank you for your suggestion. We have now written this as a lack of physical activity. Please see line 44 now.
Comment: line 51-53 - sentence should be write differently, it is hard to understand now
Response: Thank you for your suggestion. We have now rephrased the sentence. Please see page 2 (lines 52-53).
Comment: lines 65-69 - the references are in different style (Name&Name, 2020) than it s required by journal and in other paragraphs ([1]). Same in line 277
Response: Thank you for pointing it out. Referencing style is now consistent for all references according to the journals’ format.
Comment: table 1 - i belive that Authors wanted to use "<" sign in 3rd line, they use the opposite one
Response: Thank you for pointing it out. We have now used the "<" sign in the 3rd line of Table 1.
Comment: In my opinion, the article have to many flaws for current state. It should be at last rewritten and renamed, where more attention will be paid to material and methods section and more information for the above will be given.
Response: The article hugely benefited from the constructive comments and suggestions made by the reviewers. As such we have extensively revised the article. The sentences are rewritten where the information was not clear as can be seen in the track change. Additional information was added and described in greater detail in the Material and methods section.
Reviewer 3 Report
This is a review of the manuscript “Nexus between Physical Activity and Diet in Ghana”, submitted to the International Journal of Environmental Research and Public Health. The manuscript is well written and is interesting. The value of the work is in its sample, given that the context and population used have been little explored regarding the weight status and associated determinants. Nevertheless, I present below some (few) minor comments/suggestions.
Best,
DR
---------------
Abstract:
I would like to see some information about the sample in the aim. Is the analysis in the adult population? Does it include the elderly? Or just mention the age range.
Results:
In page 7, line 201, the authors report “forty five percent” while in Figure 1 the value is 46%. Please correct it.
Discussion:
In page 8, line 256-260, it is convenient to present some information about the comparative studies. Were they also carried out in Ghana or in another country with similar characteristics? It is hard to put these findings in perspective when they go along with some studies but contradicts others, but nothing is known about those previous findings. This is true for other parts of the discussion (e.g., line 266 and 271, page 9).
Page 9, line 302: please correct the word “corelates”
Author Response
Reviewer 3 comments:
Comment: In Abstract, I would like to see some information about the sample in the aim. Is the analysis in the adult population? Does it include the elderly? Or just mention the age range.
Response: Thank you for your suggestion. We have now added information about the sample in the aim (lines 9-11).
Comment: In Results, In page 7, line 201, the authors report “forty five percent” while in Figure 1 the value is 46%. Please correct it.
Response: We have now corrected the value as ‘Forty-six percent’. Please see page7, line 234 now.
Comment: In Discussion: In page 8, line 256-260, it is convenient to present some information about the comparative studies. Were they also carried out in Ghana or in another country with similar characteristics? It is hard to put these findings in perspective when they go along with some studies but contradicts others, but nothing is known about those previous findings. This is true for other parts of the discussion (e.g., line 266 and 271, page 9).
Response: Thank you for this useful suggestion. We have now presented adequate information about the comparative studies that we have cited in the discussion section in page 9 and 10.
Comment: Page 9, line 302: please correct the word “corelates”
Response: We have now corrected the word to “correlates”. Please see page 11.
Reviewer 4 Report
It was an interesting topic looking into physical activity and diet.
Title: and overall: May needs to change as my comments on the analysis part.
Abstract: Please interpret the results and not mention the correlation value only. What is the strength?
Materials and Methods:
- Please describe in detail the questionnaire in view of the item, validity.
- Do you calculate the minimum sample size?
- Sampling methods-please explain in detail. What number do you use for systematic sampling? How do you do random?
- Missing data-why you did not use multiple imputations from the model? suggest to use this and do sensitivity analysis comparing with and without missing data. May put the findings as supplementary.
Results:
- Suggest analysing the subgroup of people with and without for both physical activity and diet (Yes and no for both.). Please do extensive analysis to see an overview of this nexus.
Discussion and conclusion: May change as the changes in data analysis and results section.
Author Response
Reviewer 4 comments:
Comment: Title: and overall: May needs to change as my comments on the analysis part.
Response: Not applicable
Comment: Abstract: Please interpret the results and not mention the correlation value only. What is the strength?
Response: Thank you for this useful suggestion. We have now amended this in the abstract (lines 14-15).
Comment: Please describe in detail the questionnaire in view of the item, validity.
Response: Thank you for the suggestions. As suggested, we have now described in detail the questionnaire as well as its validity on page 3 in the ‘Design and validation of the questionnaire’ heading.
Comment: Do you calculate the minimum sample size?
Response: Yes, we calculated the minimum sample size. The minimum sample required per study site was estimated using the population size with a 95 % confidence interval and a 3% margin of error. Please see details on page 2 (lines 93-98).
Comment: Sampling methods-please explain in detail. What number do you use for systematic sampling? How do you do random?
Response: As suggested, we have now explained sampling methods in detail on pages 2 and 3, lines 87-110.
Comment: Missing data-why you did not use multiple imputations from the model? suggest to use this and do sensitivity analysis comparing with and without missing data. May put the findings as supplementary.
Response: Thank you for the suggestion to use multiple imputations to deal with missing data. We had already explained it clearly on page 4 (lines 180-184).
Comment: Suggest analysing the subgroup of people with and without for both physical activity and diet (Yes and no for both.). Please do extensive analysis to see an overview of this nexus.
Response: Thank you for your suggestion to do a subgroup analysis. However, we don’t understand what you mean by the subgroup of people with and without both physical activity and diet. Can you please explain it clearly?
Comment: Discussion and conclusion: May change as the changes in data analysis and results section.
Response: Not applicable as there is no change in the findings of the study after considering the reviewers’ comments.
Round 2
Reviewer 2 Report
The authors adjust to my comment in the response and add valuble text inside the manuscript.
The method section looks a lot better now. The technical errors were eliminated, the discussion are more careful and the introduction have necessary background (specialy about ethnicity).
I have few more comments:
- About qualification to healthy/unhealthy diet. As you said in sentences (134-139) "To qualify as eating a healthy diet, participants were asked “Thinking about the food that you eat; I would like you to tell me how often you usually eat the following foods: fruit or vegetables including fresh, frozen, dried, tinned and pure fruit juice?” Those responding at least ‘once every day’ to this question were categorized as eating a healthy diet. Those responding to other options were classified as eating an unhealthy diet. " I dont think that this is enought to qualify as healthy/unhealthy diet, its just single behavior - if respondents eat everyday hamburger from McDonald with tomato inside, they in fact eat vegetable every day and we wont consider it as healthy diet. I propose to rename healthy/unhealthy diet to eating behaviours or something like that.
- you should add information to limitation section, that qualification to healthy/unhealthy diet as well as physicaly active/not physicaly active was made by single question - some of readers may find confused about that.
- Im still not sure if title is correct, however you add some of text about this topic in discussion section, so i won`t be standing against. Please consider, however, adjusting title for the whole article.
Author Response
Comment: About qualification to healthy/unhealthy diet. As you said in sentences (134-139) "To qualify as eating a healthy diet, participants were asked “Thinking about the food that you eat; I would like you to tell me how often you usually eat the following foods: fruit or vegetables including fresh, frozen, dried, tinned and pure fruit juice?” Those responding at least ‘once every day’ to this question were categorized as eating a healthy diet. Those responding to other options were classified as eating an unhealthy diet." I don't think that this is enough to qualify as healthy/unhealthy diet, its just single behavior - if respondents eat everyday hamburger from McDonald with tomato inside, they in fact eat vegetable every day and we wont consider it as healthy diet. I propose to rename healthy/unhealthy diet to eating behaviours or something like that.
Response: Thank you very much for this constructive suggestion. We have now renamed healthy/unhealthy diets as healthy/unhealthy dietary habits. Please see page 3 (lines 140-141).
Comment: You should add information to limitation section, that qualification to healthy/unhealthy diet as well as physicaly active/not physicaly active was made by single question - some of readers may find confused about that.
Response: Thank you for your suggestion. We have now updated the limitation section of the article. Please see page 10 (lines 349-352). Qualification to be physically active/inactive was not based on a single question but a series of questions, e.g., name of the activity, duration, and intensity. Please see page 3 (lines 123-129).
Comment: I’m still not sure if title is correct, however you add some of text about this topic in discussion section, so I won’t be standing against. Please consider, however, adjusting title for the whole article.
Response: Thank you for the suggestion to improve the title. We have now amended the title. Please see page 1.
Reviewer 4 Report
Well written manuscript and justify all the comments by reviewers.
Author Response
Comments: Well written manuscript and justify all the comments by reviewers.
Response: Thank you. The article hugely benefited from the constructive comments and suggestions made by the reviewers.